# Microwave-Assisted Synthesis of SPION-Reduced Graphene Oxide Hybrids for Magnetic Resonance Imaging (MRI)

**DOI:** 10.3390/nano9101364

**Published:** 2019-09-24

**Authors:** Marina Llenas, Stefania Sandoval, Pedro M. Costa, Judith Oró-Solé, Silvia Lope-Piedrafita, Belén Ballesteros, Khuloud T. Al-Jamal, Gerard Tobias

**Affiliations:** 1Institut de Ciència de Materials de Barcelona (ICMAB-CSIC), Campus de la UAB, 08193 Bellaterra (Barcelona), Spain; mllenas@icmab.es (M.L.); oro@icmab.es (J.O.-S.); 2Institute of Pharmaceutical Science, King’s College London, London SE1 9NH, UK; pedrocosta24@gmail.com; 3Servei de Ressonància Magnètica Nuclear, Universitat Autònoma de Barcelona, Campus UAB, 08193 Bellaterra (Barcelona), Spain; silvia.lope@uab.es; 4Centro de Investigación Biomédica en Red-Bioingeniería, Biomateriales y Nanomedicina (CIBER-BBN), Universitat Autònoma de Barcelona, Campus UAB, 08193 Bellaterra (Barcelona), Spain; 5Catalan Institute of Nanoscience and Nanotechnology (ICN2), CSIC and the Barcelona Institute of Science and Technology, Campus UAB, 08193 Bellaterra (Barcelona), Spain; belen.ballesteros@icn2.cat

**Keywords:** contrast agents, biomedical imaging, ultrasmall superparamagnetic iron oxide nanoparticles

## Abstract

Magnetic resonance imaging (MRI) is a useful tool for disease diagnosis and treatment monitoring. Superparamagnetic iron oxide nanoparticles (SPION) show good performance as transverse relaxation (T_2_) contrast agents, thus facilitating the interpretation of the acquired images. Attachment of SPION onto nanocarriers prevents their agglomeration, improving the circulation time and efficiency. Graphene derivatives, such as graphene oxide (GO) and reduced graphene oxide (RGO), are appealing nanocarriers since they have both high surface area and functional moieties that make them ideal substrates for the attachment of nanoparticles. We have employed a fast, simple and environmentally friendly microwave-assisted approach for the synthesis of SPION-RGO hybrids. Different iron precursor/GO ratios were used leading to SPION, with a median diameter of 7.1 nm, homogeneously distributed along the RGO surface. Good relaxivity (r_2_*) values were obtained in MRI studies and no significant toxicity was detected within in vitro tests following GL261 glioma and J774 macrophage-like cells for 24 h with SPION-RGO, demonstrating the applicability of the hybrids as T_2_-weighted MRI contrast agents.

## 1. Introduction

Magnetic resonance imaging (MRI) is one of the most useful diagnostic tools. MRI avoids the use of radiation and provides exhaustive anatomical information with high spatial and temporal resolution. MRI has a high capacity to detect soft tissues and in the clinical setting is employed, among others, for the detection of tumours and imaging of the nervous system [1,2,3,4]. The diagnostic value of MRI can be further expanded by using exogenous contrast agents that improve the resolution of the technique allowing a better interpretation of the acquired images. These agents enhance image contrast by decreasing the longitudinal (T_1_) and transverse (T_2_) relaxation time of nearby water protons. T_1_ contrast agents are composed of paramagnetic metal ions, which have a permanent magnetic moment due to unpaired electrons that stimulate the energy transfer from nuclear spins to environment, thus reducing T_1_ relaxation. The mechanism of preferential T_2_/T_2_* shortening by the so-called T_2_ contrast agents is derived from bulk susceptibility effects and distortions of the local magnetic field [2,3,4,5,6,7,8]. Different studies have been reported on the development of contrast agents based on inorganic nanoparticles [9,10,11,12,13]. Gd^3+^ based compounds have been proposed as high-performance T_1_ contrast agents. However, their use can lead to heart failure, renal toxicity, deposition into the skin, kidneys and brain. The high toxic response, as consequence for example of the competitive inhibition of biological processes requiring Ca^2+^, increases the importance of exploring new alternative compounds [14,15,16].

Superparamagnetic iron oxide nanoparticles (SPION) have received great attention as MRI contrast agents because they dramatically shorten T_2_. More recently, efforts have been focused on their use as T_1_ contrast agents when they have small sizes, around 5 nm. SPION also show high biocompatibility, bioactivity, and conformation stability, becoming an interesting alternative to gadolinium-based T_1_ contrast agents [15,17,18,19,20,21]. Some studies reported that the aggregation of SPION enhances relaxivity (r_2_) and increases MRI contrast [22,23,24]. Inside the body, uncontrolled aggregation is not desired, as it causes nanoparticle precipitation, thus reducing their stability, biocompatibility, circulation time and efficiency [22,23,24,25,26,27].

To overcome these limitations, SPION have been combined with different stable supports, such as carbon materials, in order to prevent the agglomeration in the organism, enable a good dispersion and improve their circulation time and efficiency [19,22,23,24,28]. The combination also takes advantage of the intrinsic properties of both materials [27], and an MRI contrast enhancement has also been observed by the interaction of magnetic nanoparticles loaded onto neighbouring nanocarriers [29].

Graphene is a two-dimensional material that presents a single layer of sp^2^-hybridized carbon atoms in a hexagonal lattice. It presents unique physical, electrical, chemical, and mechanical properties that, along with good biocompatibility, make graphene and their derivatives attractive for a myriad of biomedical applications, such as biosensors, drug delivery, or bioimaging [30,31,32,33,34,35,36,37]. Graphene oxide (GO) is a graphene derivative with a high content of oxygen-containing functional groups, while reduced graphene oxide (RGO) is obtained by reduction of GO, thus decreasing the number of functional moieties. These materials present high specific surface area and oxygen functionalities, making them ideal substrates for the immobilization of inorganic nanoparticles. The resulting hybrid material benefits from the properties of both constituent compounds [37,38,39,40,41]. Thus, the decoration of carbon-based materials such as GO or RGO with SPION is an appealing strategy for the development of T_2_-weighted magnetic resonance contrast agents [41,42,43].

Different methods have been proposed in order to obtain decoration of GO or RGO in situ with iron oxide nanoparticles, such as co-precipitation [44,45], hydrothermal [23], or solvothermal approaches [42,46,47]. Some other strategies have been based on the decoration of graphene compounds by an ex situ approach: synthetizing first the nanoparticles and then attaching them onto the graphene surface [14]. In the present work, we report the in situ formation of superparamagnetic iron oxide nanoparticles-reduced graphene oxide (SPION-RGO) hybrids using iron(III) acetylacetonate (Fe(acac)_3_) as precursor and benzyl alcohol as solvent under microwave treatment. Microwave-assisted synthesis is based on the use of microwave irradiation to heat the reaction mixture. Microwave radiation is an electromagnetic energy source that can be divided into electric and magnetic components. The electric component causes heating by two main mechanisms: dipolar polarization and ionic conduction. In the first case, the dipoles present in the sample align in the direction of the applied electric field upon microwave irradiation, further attempting to follow the oscillations of this field. Thus, the heat is consequently generated from molecular friction and dielectric loss. In the case of ionic conduction, the charged particles or ions oscillate back and forth under the influence of the microwave field, colliding with their neighbouring molecules or atoms, which causes rising of the temperature due to the collisions and frictions [48,49,50]. This approach presents some interesting advantages such as the reduction of synthesis time and the homogenous heat supply [51,52]. During the microwave treatment, the functional groups of GO act as heterogeneous active sites for the nucleation of the metal oxide [53,54]. It has been previously shown that the properties of the SPION-RGO hybrid could be tuned by a variety of parameters such as substrate pre-treatments, metal salt loading or method of mixing [53,54]. Microwave-assisted synthesis has been previously used to prepare SPION-RGO/GO composites obtaining uniform nanoparticles coatings along the substrates and the resulting hybrid materials have been proposed for applications in the energy field, including batteries [55,56] and supercapacitors, to name some [57].

The cytotoxicity of the nanocomposites is critical in order to assess their applicability in the biomedical field [58,59]. The toxicity of RGO/GO-based nanomaterials depends on the synthesis method, size and morphology [59] and some previous studies have already tested the viability of SPION-RGO/GO composites synthetized using different methods, such as co-precipitation [45], hydrothermal [23], solvothermal [42] or ex situ synthesis [14,24,60]. Herein we have assessed the stability of the hybrids in serum, evaluated their cytotoxicity and performed phantom MRI studies to explore the potential of the microwave-assisted approach to develop these contrast agents for MRI.

## 2. Materials and Methods

### 2.1. Reagents

Graphite powder (20 µm), Benzyl alcohol (99.8%), iron(III) acetylacetonate (>99%), NaNO_3_ (>99%) and KMnO_4_ (>99%) were purchased from Sigma-Aldrich. Ethanol (99.9%, Panreac química, SLU), H_2_SO_4_ (95–97%, Scharlab), and H_2_O_2_ (30%, Panreac química, SLU).

Human serum for the stability test was purchased from Sigma-Aldrich and for the in vitro assays, Fetal bovine serum (FBS) was obtained from First-Link Ltd. (Wolverhampton, UK), Phosphate Buffered Saline (PBS), advanced RPMI (Roswell Park Memorial Institute) 1640, Penicillin-Streptomycin 100X, 0.05% trypsin-EDTA (1X) with Phenol Red and GlutaMAX™ Supplement were obtained from Life Technologies (UK). Pluronic F-127 was acquired from Sigma-Aldrich (UK) and the CytoTox 96^®^ Non-Radioactive Cytotoxicity Assay was obtained from Promega Corporation, UK.

### 2.2. Synthesis of Graphene Oxide

Graphene oxide was prepared via a modified Hummers’ method following a previously reported protocol [61]. For this purpose, 5 g of graphite powder were mixed with 115 mL of H_2_SO_4_ and 2.5 g of NaNO_3_, keeping the mixture down to 0 °C. Afterwards, 15 g of KMnO_4_ were added slowly while the temperature was carefully controlled to keep the reaction below 20 °C. The reaction mixture was warmed to 35 °C and maintained for 30 min under continuous stirring. Then, 230 mL of water were added, and the temperature was raised to 98 °C and stirred for 2 h. Afterwards, 700 mL of water and 5 mL of 30% H_2_O_2_ were added slowly. When the mixture reached the room temperature the content was repeatedly centrifuged and washed with distilled water until the pH of the solution was neutral. Finally, the resulting solid was dried at 60 °C.

### 2.3. Decoration of RGO with SPION

SPION-RGO samples were synthesized using a microwave-assisted method adapted from Roig et al. on the synthesis of water-dispersible iron oxide nanoparticles [62]. For this purpose, GO (15 mg) was dispersed in 5 mL of benzyl alcohol using an ultrasound bath, and the dispersion was subsequently placed in a 10 mL microwave tube and mixed with 30 mg of iron(III) acetylacetonate (Fe(acac)_3_). The solution was mixed for two minutes, placed inside the microwave oven (Discover Explorer Hybrid, CEM) and the two-stage program registered in Table 1 was performed. The resulting black suspension was filtrated and washed several times with distilled water, and the powder was dried at 60 °C. Additional samples using 25 and 35 mg of (Fe(acac)_3_), while keeping the amount of GO constant at 15 mg, were prepared to study the differences when the amount of precursor is changed.

### 2.4. Characterization

The morphology and size of the RGO sheets, as well as the distribution and size of SPION on the RGO surface, were evaluated by transmission electron microscopy (TEM) using a JEOL Jem 1210 electron microscope operating at 120 kV and scanning electron microscopy (SEM) using a QUANTA FEI 200 FEG-ESEM. The size distribution of SPION was determined from 596 NPs using TEM images. The diameter of the particles was measured using ImageJ software, along with an additional plugin that allows discerning between aggregated particles by approaching the NP shape to a circumferential model. The area of RGO was determined from 145 sheets using SEM images, multiplying the measured length and width for each of the analysed sheets. High-resolution transmission electron microscopy (HRTEM) (FEI Tecnai F20 S/TEM) and Energy Filtered TEM (EFTEM) images were acquired on a FEI Tecnai F20 S/TEM coupled to a Gatan Imaging Filter (GIF) Quantum SE 963 fitted with a 2k × 2k CCD camera. Energy-filtered images were acquired over the carbon K edge (onset at 284 eV) and the iron L edge (onset at 708 eV) with a 20–40 eV window using the three-window method. Images were acquired with 5 s acquisition and binned to render 512 × 512 images. Colour mixing was performed with a dedicated script in Gatan Digital Micrograph software. Samples were prepared by dispersing a small amount of powder in ethanol and placed dropwise on a lacey carbon grid. Energy-dispersive X-ray spectra were obtained with FEI Tecnai F20 S/TEM. X-ray Photoelectron Spectroscopy (XPS) was performed using a hemispherical energy analyser (PHOIBOS 150, SPECS). Ethanol dispersions containing the sample were prepared, placed dropwise onto 5 × 5 mm silicon wafers and subsequently evaporated at room temperature. Thermogravimetric analyses were performed on a Netzsch instrument, model STA 449 F1 Jupiter^®^, under flowing air at a heating rate of 10 °C·min^−1^. Finally, a Superconducting Quantum Interference Device (SQUID) Magnetometer (Quantum Design) with field values of ±50 kOe and temperatures of 10 K and 300 K, was used to determine the hysteresis loops of the samples, by performing the magnetic versus field loop measurements.

### 2.5. In Vitro MRI studies

^1^H-magnetic resonance imaging (MRI) studies were performed in a 7 T Bruker BioSpec 70/30 USR (Bruker BioSpin GmbH, Ettlingen, Germany) system equipped with a mini-imaging gradient set (400 mT/m) and using a linear volume coil with 72 mm inner diameter. MR data were acquired and processed on a Linux computer using Paravision 5.1 software (Bruker BioSpin GmbH, Ettlingen, Germany).

For relaxivity measurements, phantoms containing SPION-RGO at five different concentrations (0.25, 0.5, 0.75, 1 and 1.25 mM) in 1% agarose were prepared. MR images were obtained from 3 mm slice thickness coronal sections with a field of view (FOV) of 8 × 4 cm^2^ and a matrix size = 128 × 128. Longitudinal relaxation times (T_1_) were measured using a spin-echo sequence with variable repetition time (TR = 300, 500, 700, 1000, 1300, 1700, 2000, 2600, 3500, and 5000 ms), and echo time (TE) = 7.5 ms. For transverse relaxation time (T_2_) measurements, a multi-slice multi-echo sequence was used, with TR = 3 s, and TE values between 10 and 600 ms in steps of 10 ms. T_2_* values were measured with a fast-low angle shot (FLASH) sequence, with flip angle = 20°, TR = 700 ms, and variable TE values (3, 4, 6, 8, and 10 ms). Data were fitted to exponential curves to obtained T_1_, T_2_, and T_2_* relaxation times, respectively. Longitudinal and transverse relaxivity values, r_1_, r_2_, and r_2_*, were obtained as the slope of the linear regression of the relaxation rates (R, as the inverse of the relaxation times) versus metal concentration.

### 2.6. In Vitro Stability Assay

To test the stability of the composites in the biological medium, SPION-RGO (prepared with 30 mg of Fe(acac)_3_) were incubated in human serum. Samples were incubated up to 24 h at 37 °C. After the incubation, the collected hybrids were analysed by TEM in order to assess their structural stability.

### 2.7. In Vitro Modified Lactate Dehydrogenase (LDH) Assay

GL261 mouse glioma and J774 mouse macrophage-like cell lines were maintained in Advanced RPMI containing 4.5 g·L^−1^ glucose, supplemented with 10% heat-inactivated FBS, 100 U·mL^−1^ penicillin, 100 µg·mL^−1^ streptomycin and 2 mM GlutaMax, at 37 °C under a humidified atmosphere containing 5% CO_2_. Cells were seeded onto a 96-well plate at a density of 1 × 10^4^ (GL261) or 8 × 10^3^ (J774) cells per well. Twenty-four hours after plating, SPION-RGO dispersions (in 1% Pluronic F-127) were added to the cells at a final concentration of 10, 50 or 100 μg·mL^−1^. At the experiment endpoint (24 or 72 h post-incubation), the cells were washed three times with PBS and lysed by incubation for 1 h at 37 °C with 0.9% Triton X-100 (*v*/*v* in fresh phenol-free medium). The cell lysates were subsequently centrifuged for 2 h at 4000 rpm (Eppendorf 5810R, Germany) and the supernatant was recovered for determination of LDH activity using the CytoTox 96^®^ assay (Promega Corporation, USA). The absorbance at 490 nm was measured in a FLUOstar Omega microplate reader (BMG Labtech, Ortenberg, Germany). Cell viability (presented as a percentage to control untreated cells) was calculated as follows:$${\%}_{viability}=\frac{A_{490}\;treated\;cells-A_{490}\;negative\;control}{A_{490}\;untreated\;cells-A_{490}\;negative\;control}\cdot 100$$
where “A_490_ of negative control” represents the absorbance at 490 nm of the lysis solution.

## 3. Results and Discussion

Reduced graphene oxide decorated with superparamagnetic iron oxide nanoparticles (SPION-RGO) was prepared using a microwave-assisted method. This approach involves the in situ synthesis of SPION by decomposition of iron(III) acetylacetonate and their simultaneous attachment onto the surface of the resulting RGO. For this purpose, a given amount of iron precursor (30 mg) was mixed with GO and microwave treated following the synthesis steps detailed in Table 1. Graphene oxide presents a high absorption of microwave radiation and acts as the main microwave absorber in the system. Therefore, it can be selectively heated, leading to the nucleation of the metal oxide onto its surface [54]. During the microwave-assisted synthesis, the functional groups from GO chemically bond and stabilize the metal oxide nanoparticles [63]. At the same time, benzyl alcohol acts as a reactant for the synthesis of metal oxide nanoparticles and also as reducing agent, leading to the partial reduction of GO [54]. Thus, after the microwave-assisted process, the obtained material consists of RGO supporting iron oxide nanoparticles [64].

The prepared SPION-RGO composites were analysed by TEM and SEM to determine their morphology. As it can be observed in the TEM image in Figure 1a, the RGO surface is homogeneously and almost completely covered by well-defined nanoparticles (NPs). EDX analysis, shown in Figure 1b, confirms the presence of Fe from the synthesized NPs and C, arising from RGO. The median particle size of SPION, as determined from *ca.* 600 NPs, turned out to be 7.1 nm (see the corresponding histogram in Figure 1c). Some previous publications have already reported the synthesis of similar composites using a microwave-assisted method. Baek et al. [54] obtained a good uniform deposition of iron oxide nanoparticles along the substrate and with no free particles present, while Liu et al. [55] reported composites with relatively uniform and dispersed nanoparticles with a size around 4–8 nm. Garino et al. [65] showed that nucleation of different density and sizes of iron oxide nanoparticles (*ca.* 5–10 nm and 30–80 nm) could be achieved by tuning the loading of iron precursor. Thus, the microwave-assisted synthesis has been demonstrated to be a versatile approach, since it can be used to obtain iron oxide on RGO/GO composites with different nanoparticle sizes and loadings.

SEM was employed to determine the dimensions of the RGO sheets. The area of RGO was determined by multiplying the measured length and width for each of the analysed sheets. The resulting histogram (Figure 1d) reveals a broad size distribution with a 9.2 µm^2^ median area. Representative SEM images are shown in Figure 1e and Appendix A. Statistical data and box plot analysis for both SPION and RGO are included in Appendix A and Appendix A respectively. Finally, the homogeneous distribution of the SPION along the RGO surface was confirmed by EFTEM compositional maps of carbon (coloured in red) and iron (coloured in green) (see Figure 2 and Appendix A).

A HRTEM image of SPION-RGO is presented in Figure 3a, where several nanoparticles covering the RGO sheets can be seen. The selected area electron diffraction (SAED) analyses (inset) shows the characteristic pattern of the graphene hexagonal structure ((001) plane) with the observed diffraction spots (marked with green circles), corresponding to the (100) and (110) reflections (d spacing = 2.13 Å and 1.23 Å, respectively). The spots overlap with concentric diffraction rings, resulting from diffraction of SPION that cover the RGO in different orientations. Lattice fringes are clearly visible for some nanoparticles in the HRTEM image. The interplanar distances determined from the SAED pattern are in good agreement with those of maghemite (γ-Fe_2_O_3_) and magnetite (Fe_3_O_4_) structures of iron oxide [28]. Similarly, the acquired X-ray diffraction pattern (Figure 3b) does not allow discerning between both crystalline structures. Thus, the diffraction peaks can be indistinctly attributed to both maghemite and magnetite. X-ray photoelectron spectroscopy (XPS) was next performed and the high-resolution Fe 2p spectrum of the SPION-RGO is shown in Figure 3c. The energy separation between the satellite peak and the main peak (ΔEs) in the spectrum is in good agreement with the presence of Fe^3+^ (ΔEs = 8.3 eV) [66]. Thus, it can be inferred that the synthesized nanoparticles correspond to maghemite (γ-Fe_2_O_3_).

Having confirmed the successful decoration of RGO with iron oxide NPs, we next investigated how the amount of iron precursor, employed for the synthesis of the hybrids, influences the final loading with NPs and the magnetic properties of the resulting hybrids. For this purpose, the amount of GO was kept constant (15 mg) while 25, 30, or 35 mg of iron(III) acetylacetonate were used for the syntheses. TEM inspection of the hybrids prepared using different iron precursor/GO ratios confirms the successful decoration of the RGO sheets with NPs (Appendix A). Thermogravimetric analyses (TGA) were employed to quantitatively determine the final loading of SPION attached to RGO as well as the thermal stabilities of the analysed materials. Three well defined thermal events are observed during the TGA of GO under flowing air (Appendix A). An initial weight loss up to *ca.* 100 °C is observed and it is attributed to absorbed water. At *ca.* 220 °C elimination of O-based aliphatic moieties starts to occur and the final weight loss accounts for the removal of the more stable functional groups along with the total combustion of the graphitic network (above 500 °C). The TGA of the SPION-RGO hybrids suggests the partial reduction of the material (formation of RGO). On the one hand, less atmospheric water is absorbed on the sample, as inferred from the dramatic reduction of the initial thermal event (up to *ca.* 100 °C in GO), and on the other hand, a large decrease in the weight loss associated to aliphatic moieties is also observed. The complete oxidation of the hybrids takes place at lower temperature than GO, with a *ca.* 150 °C decrease in the onset temperature of combustion. It is well known that the presence of inorganic material might decrease the onset temperature of combustion of carbon nanomaterials [67,68]. Taking into account that the NPs are γ-Fe_2_O_3_, the number of NPs loaded in each of the hybrids can be directly determined from the TGA residue. Thus, SPION-RGO samples containing 28.9 wt. %, 33.6 wt. %, and 37.3 wt. % of γ-Fe_2_O_3_ are obtained by using 25 mg, 30 mg and 35 mg of Fe(acac)_3_, respectively. As expected, an increase in the amount of iron precursor leads to a higher degree of loading with NPs.

The magnetic properties of the different hybrids were next evaluated at 10 K and 300 K (Figure 4a and Appendix A, respectively). In both cases, no remnant magnetization was observed in the absence of an applied magnetic field, in agreement with the typical behaviour of superparamagnetic materials [17]. The saturation magnetization (M_s_) of the samples is plotted against the iron oxide content determined by TGA (Figure 4b). As it can be observed, the higher the amount of iron oxide, the higher the magnetization at a given applied field. Taking into account that the formation of maghemite NPs with similar sizes would be expected in the three prepared samples, the magnetic response can be mainly attributed to the amount of loading of the iron oxide nanoparticles.

Having confirmed the superparamagnetic behaviour of the SPION-RGO, hereinafter the sample with an intermediate loading (33.6 wt. %) was employed for testing the properties of the synthesized nanohybrid. This sample was chosen because we have observed a good MRI response on a previous study using carbon nanotubes with a similar loading of magnetic nanoparticles [29].

MRI T_2_ and T_2_* weighted images of SPION-RGO at different concentrations are shown in Figure 5a,b respectively, demonstrating contrast variations for the different Fe concentration. Fe content reduces transverse relaxation times by enhancing the loss of spin phase coherence. Thus, in T_2_ and T_2_* weighted images, low Fe concentration regions appear bright, whereas those with higher Fe concentrations appear darker in the images. Therefore, T_2_ and T_2_* contrast enhancement are directly dependent on the Fe concentration, which is in agreement with previous reports [42,69].

The contrast agent efficiency in MRI is given by its relaxivity, which describes how much the relaxation rate (the inverse of the relaxation time) increases due to the contrast agent concentration. Figure 5 shows the relaxivity curves by plotting the relaxation rates, R_2_ (c) and R_2_* (d), for each Fe concentration. Both transverse relaxation rates R_2_ and R_2_* reflect the spin-spin relaxation process, however, R_2_* takes also into account the losses of phase coherence due to magnetic field inhomogeneity. Anisotropic SPION-RGO composites induce stronger field inhomogeneity exacerbating R_2_* effect.

The transverse relaxivity (r_2_) of SPION-RGO was measured to be 9.75 mM^−1^·s^−1^ and the r_2_* is 392.9 mM^−1^·s^−1^. The SPION-RGO r_2_ value is lower than clinically approved SPION like Endorem or Resovist with r_2_ values above 100 mM^−1^·s^−1^ [70], probably due to well-known size-dependent T_2_ characteristics since these have bigger hydrodynamic diameters of about 150 nm (Endorem) and 60 nm (Resovist) [7,70]. However, the r_2_* value of SPION-RGO is similar to the value of Endorem (325 mM^−1^·s^−1^) showing the viability of SPION-RGO as contrast agent for MRI studies [71]. The observed high r_2_* value could be attributed to the anisotropy of the SPION-RGO and the strong magnetic field inhomogeneity that they produce.

Longitudinal relaxation (R_1_) results were observed to be too low to have applicability as a T_1_ contrast agent (Appendix A). Some previous studies found that ultrasmall particles, with diameters around 5 nm, presented good applicability as T_1_ contrast agents [15,20,21]. Despite the size of the present nanoparticles is quite small (around 7 nm), no appreciable signal was observed in this case. The fact that the present composites do not present high r_1_ values might be due to the reduced accessibility of the water molecules to SPION that were partially obstructed in our case by RGO or maybe the size is not small enough to observe the enhancement in r_1_ values.

As potential material for biomedical applications, the stability of the SPION-RGO hybrid in the biological medium was tested. The sample was immersed in human serum at 37 °C during 24 h. The sample was then collected and analysed by TEM. As can be seen in Figure 6, the hybrid showed good stability in a biological medium and no significant variations in the morphology and SPION coverage were observed.

The toxicity of SPION-RGO in brain cancer-derived (GL261) and macrophage-like (J774) mouse cell lines were independently assessed using a modified version of the traditional LDH assay, due to interference of carbon compounds with the assay components [72]. As shown in Figure 7, significant toxicity could not be detected when the GL261 and J774 cells were incubated for 24 h with SPION-RGO in 1% Pluronic F-127, with only a minor reduction in cell viability found in J774 cells at both 50 and 100 μg·mL^−1^ (~22% reduction, *p* > 0.05). Similarly, a significant reduction in cell viability could not be found when J774 cells were incubated for 72 h with different concentrations of SPION-RGO (highest decrease detected for 100 μg·mL^−1^, ~21%), whereas moderate cell toxicity was detected in GL261 cells incubated for 72 h with 100 μg·mL^−1^ (~41% decrease, *p* > 0.05).

Apart from studying the cytotoxicity of the SPION-RGO composites, it is important to identify any potential cell damage associated with them. Next steps to advance in this field would include assessing the degradation of the composites, taking into account the possible dissolution of iron oxide nanoparticles, and the potential toxicity, if any, of the degradation products.

Previous studies have already investigated the degradability of SPION. For example, Nkansah et al. [73] tested the degradation of iron oxide nanoparticles mimicking the lysosome environment. They found that 100 days were needed to completely eliminate them under conditions employed in their study. Some other works also postulate that SPION can be degraded and cleared from circulation by the endogenous iron metabolic pathways, so iron released from SPION can be metabolized in the liver and then used in the formation of red blood cells or excreted via kidneys [74]. By means of MRI, the in vivo behavior of intravenously injected iron oxide nanoparticles has been assessed [75,76,77].

On the other hand, the degradability, toxicity, and elimination of graphene derivatives have also been investigated [78,79]. These studies highlight the importance of understanding the potential toxicity of nanomaterials, as well as their interaction with the immune system, which is critical for their application in the biomedical field.

## 4. Conclusions

In summary, SPION has been synthesized in situ using GO as support. The employed microwave-assisted synthesis also leads to the simultaneous reduction of GO, thus resulting in the formation of SPION-RGO hybrids. Combined SAED, XRD, and XPS analysis reveal the formation of maghemite (γ-Fe_2_O_3_) nanoparticles. The degree of loading can be easily controlled by changing the stoichiometric ratio of GO and the iron precursor during the synthesis of the hybrids, which in turn determines the magnetic behaviour of the SPION-RGO. The prepared hybrids present good stability in human serum and cell viability at 24 h with GL261 and J774 cell lines. Relaxivity (r_2_*) measurements revealed a similar value as Endorem, a commercial T_2_ contrast agent, thus demonstrating the applicability of the SPION-RGO composites as T_2_-weighted MRI contrast agents.

## Figures and Tables

**Figure 1 nanomaterials-09-01364-f001:**
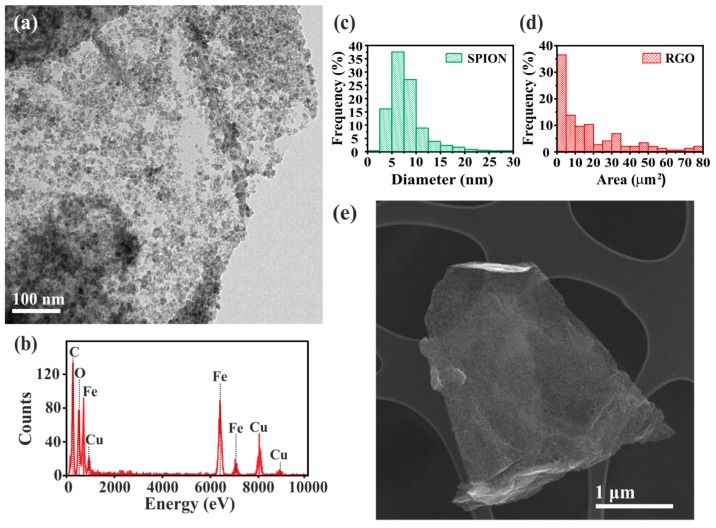
Electron microscopy characterization of the SPION-RGO composites. (**a**) TEM image of SPION-RGO sample prepared by microwave-assisted decomposition of iron (III) acetylacetonate in the presence of graphene oxide. (**b**) EDX analysis confirming the presence of Fe in the sample (from SPION) along with C (from the RGO). Cu signal arises from the support employed for the analysis. Histograms showing (**c**) the particle size distribution of the SPION decorating the RGO surface and (**d**) the area of the RGO sheets. (**e**) Representative SEM image of the SPION-RGO sample.

**Figure 2 nanomaterials-09-01364-f002:**
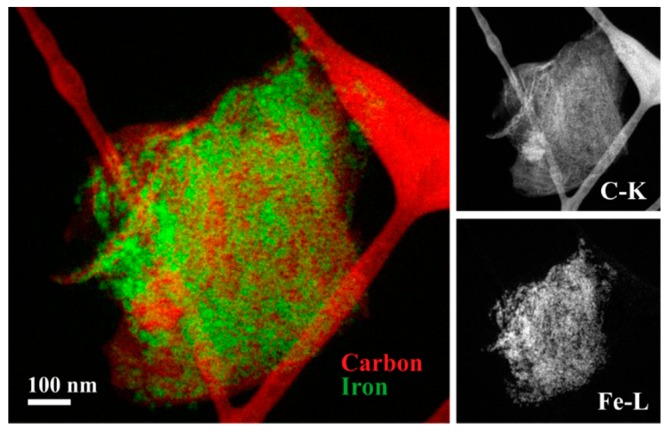
EFTEM elemental distribution maps of carbon and iron in the SPION-RGO composites.

**Figure 3 nanomaterials-09-01364-f003:**
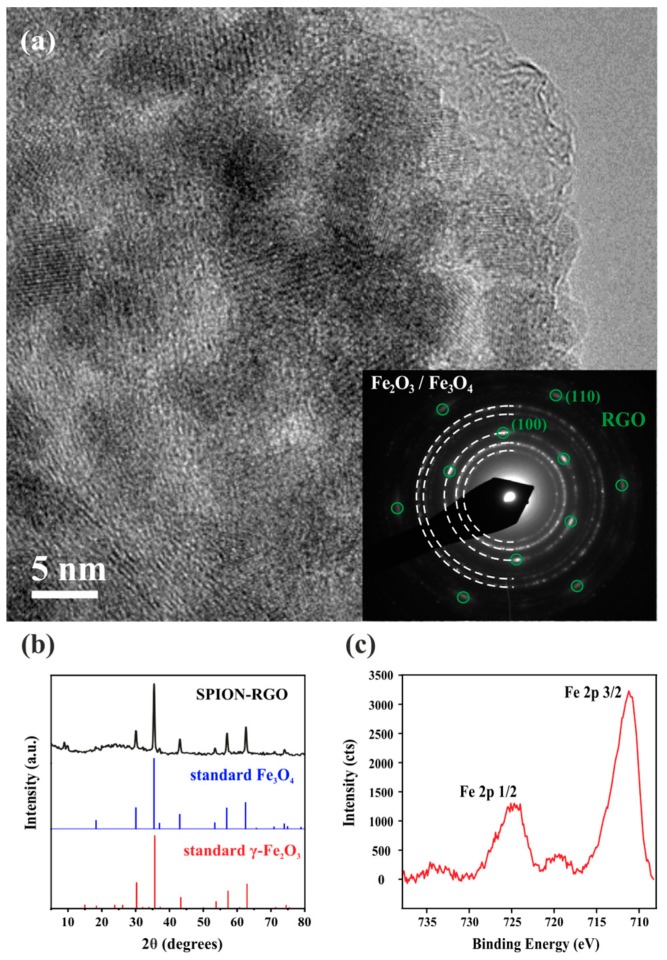
(**a**) HRTEM image of SPION-RGO and the corresponding SAED pattern (inset), (**b**) X-ray diffraction pattern of the hybrid material along with the standard diffraction patterns of Fe_3_O_4_ (PDF#: 722303) and γ-Fe_2_O_3_ (PDF#: 391346), and (**c**) high resolution Fe 2p X-ray photoelectron spectrum of SPION-RGO.

**Figure 4 nanomaterials-09-01364-f004:**
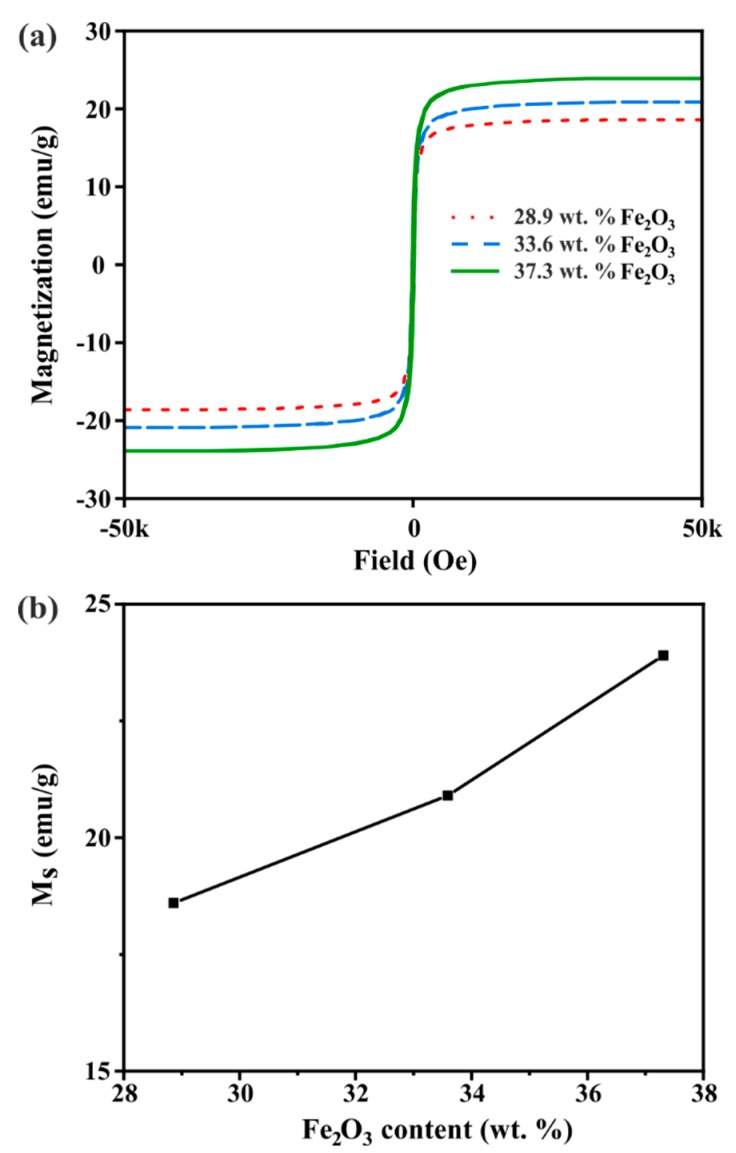
(**a**) SPION-RGO hysteresis loops (magnetization versus field) at 10 K and (**b**) saturation magnetization (M_s_) of the hybrids with respect to the loading of iron oxide NPs.

**Figure 5 nanomaterials-09-01364-f005:**
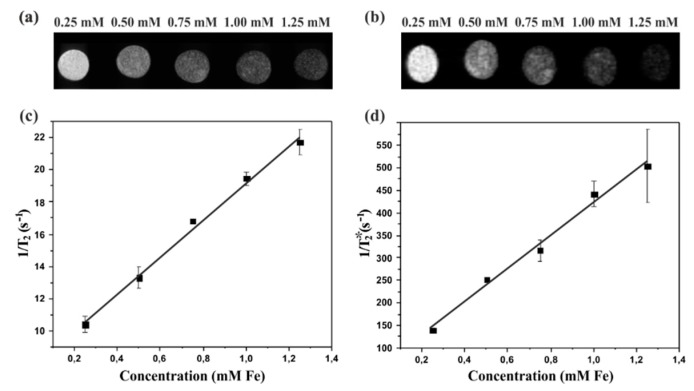
Phantom MRI studies of SPION-RGO. (**a**) T_2_-weighted and (**b**) T_2_*-weighted images of SPION-RGO at different concentrations. (**c**) R_2_ and (**d**) R_2_* relaxation rates versus Fe concentration. The relaxivity values were obtained from the slopes. Results are the mean value ± S.D. (*n* = 2).

**Figure 6 nanomaterials-09-01364-f006:**
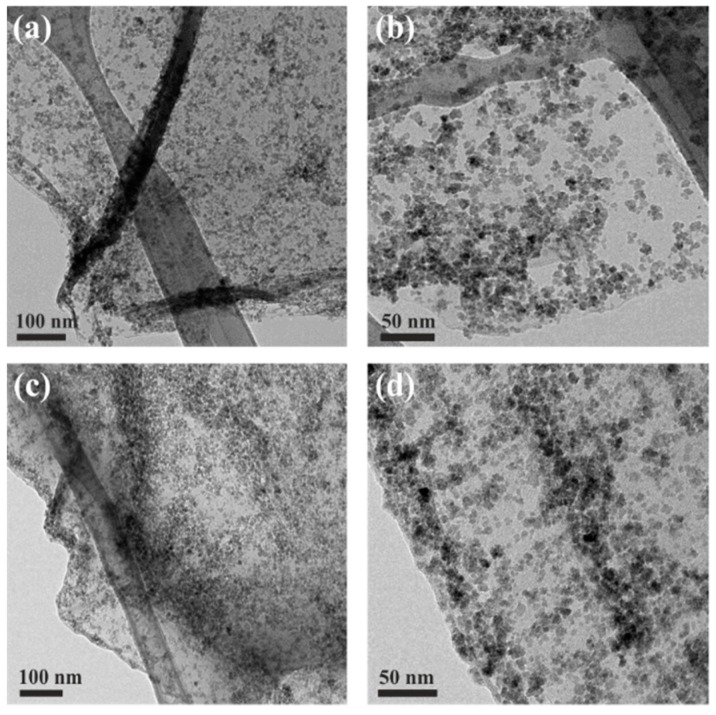
TEM images of SPION-RGO hybrids (**a**,**b**) before and (**c**,**d**) after being exposed to human serum at 37 °C for 24 h.

**Figure 7 nanomaterials-09-01364-f007:**
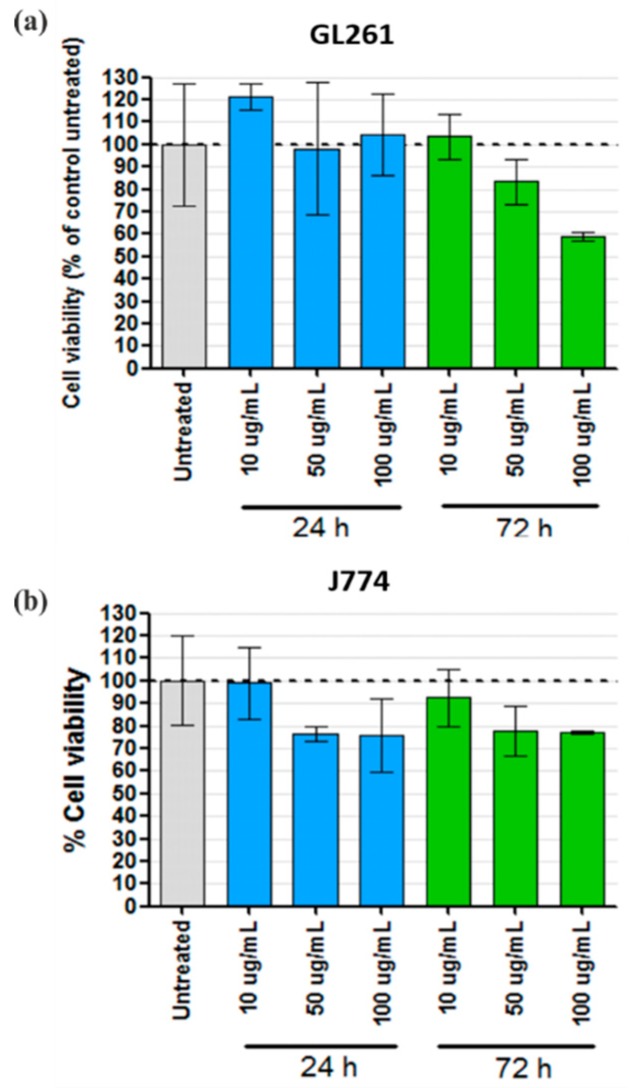
Cell toxicity of SPION-RGO assessed in GL261 glioma and J774 macrophage-like cell lines via modified LDH assay. (**a**) GL261 and (**b**) J774 cells after incubation with different concentrations of SPION-RGO for 24 or 72 h. Results (mean ± SD) are expressed as a percentage of cell viability compared to control untreated cells (two different experiments performed with triplicates).

**Table 1 nanomaterials-09-01364-t001:** Microwave conditions used for the synthesis of SPION-RGO.

Step	T (°C)	Time (Min)	Max. Power (W)	Max. Pressure (Bar)
**1**	60	5	300	12
**2**	180	10	300	12

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
