# Peer review of "Microwave-Assisted Synthesis of SPION-Reduced Graphene Oxide Hybrids for Magnetic Resonance Imaging (MRI)"

_nanomaterials, 2019, doi:10.3390/nano9101364_

Round 1
Reviewer 1 Report
Major comments:
In the manuscript, the author used the microwave to produce the nanomaterials, thus, the author needs to describe the microwave-assisted synthesis in the introduction section. As the results, we could find the SPION-RGO composites were aggregated, and how to identify the particle size and calculate the diameter with manual counting? There many publications focus on the microwave-assisted synthesis, and they could produce the uniform and dispersed nanomaterials, I suggest that the author needs to compare with other publications. In addition, the r2 relaxivity of SPION was approach 100 mM-1S-1, such as Resovist, but the r2 relaxivity of SPION-RGO was only 9.75 mM-1S-1, I suggest the author need to discuss this difference in the manuscript. For the biosafety of SPION-RGO, the author demonstrated the cell viabilities. Herein, the cell lines were cancer and macrophage, as we known these cell lines have higher tolerance for nanomaterials, I suggest the author need to apply the normal cell to identify the cytotoxicity of SPION-RGO. In addition, how about the immune response of SPION-RGO?
Reviewer 2 Report
The subject matter of this paper deals with synthesis of SPION-reduced graphene oxide hybrids for magnetic resonance imaging. The work is interesting, well organized and comprehensively described. To improve the manuscript I recommend using asterisks for significant differences in Figure 7. I also recommend discussing some recent articles:
Gonzalez-Rodriguez R, Campbell E, Naumov A. Multifunctional graphene oxide/iron oxide nanoparticles for magnetic targeted drug delivery dual magnetic resonance/fluorescence imaging and cancer sensing. PLoS One. 2019 Jun 6;14(6):e0217072. doi: 10.1371/journal.pone.0217072. eCollection 2019. Sun W, Huang S, Zhang S, Luo Q. Preparation, Characterization and Application of Multi-Mode Imaging Functional Graphene Au-Fe3O4 Magnetic Nanocomposites. Materials (Basel). 2019 Jun 19;12(12). pii: E1978. doi: 10.3390/ma12121978. Luo Y, Tang Y, Liu T, Chen Q, Zhou X, Wang N, Ma M, Cheng Y, Chen H. Engineering graphene oxide with ultrasmall SPIONs and smart drug release for cancer theranostics. Chem Commun (Camb). 2019 Feb 7;55(13):1963-1966. doi: 10.1039/c8cc09185d. Gusev A, Zakharova O, Vasyukova I, Muratov DS, Rybkin I, Bratashov D, Lapanje A, Il'inikh I, Kolesnikov E, Kuznetsov D. Effect of GO on bacterial cells: Role of the medium type and electrostatic interactions. Mater Sci Eng C Mater Biol Appl. 2019 Jun;99:275-281. doi: 10.1016/j.msec.2019.01.093. Epub 2019 Jan 22. Lin J, Chen X, Huang P. Graphene-based nanomaterials for bioimaging. Adv Drug Deliv Rev. 2016 Oct 1;105(Pt B):242-254. doi: 10.1016/j.addr.2016.05.013. Epub 2016 May 24.Author Response
Please see the attachment.

Reviewer 3 Report
The manuscript entitled “Microwave-assisted synthesis of SPION-reduced graphene oxide hybrids for magnetic resonance imaging (MRI)” has been submitted for publishing consideration. Since nanomaterials are rapidly developing nowadays, investigation on their medical applications is clearly an important topic. Taking advantages of previous report on SPION decorated on carbon nanotubes (Adv. Funct. Mater. 2014, 24, 1880–1894), the authors present in this work SPION decorated graphene oxide for MRI application.
There are some questions I am concerning about and hope that the authors could integrate their answer to the revised manuscript:
1. In previous works, Iron oxide nanoparticles with low dimensions, such as ultrathin nanowires (diameter < 3 nm) and ultrasmall nanospheres (<4 nm), have been studied as clinically-preferred T1 contrast agents. The nanoparticle size in this work is however >= 5nm, and is used as T2 contrast agent. What is the merit of this bigger size SPION and using it as T2 contrast agent (compared to the smaller one)?
2. Why does it need to be with RGO? If the purpose is to increase the dispersibility of SPION by attaching to some hydrophilic backbone, then graphene oxide is good and even better than RGO. The authors will not need to add the benzyl alcohol to reduce the GO to RGO.
3. Compared to the previous report on the SPION-MWCNTs, which one is better? And at which aspect?
4. The authors should investigate particle degradation, iron dissolution and thus the iron uptake following cell labeling as it is important aspect toward real application. (Please compare with information given in Ref: Magnetic Resonance in Medicine 73:376–389 (2015)). For example, Nkansah et al investigated the dissolution of iron over time by incubating magnetic particles in citrate buffer (mimic the lysosomal environment to which particles are exposed following endocytosis by cells) over 100 days and measuring the solubilized iron. They found that after 1 day, the microparticles lose 3% of their iron content due to dissolution, while the nanoparticles lose as much as 21% in the same time frame.
Other than those questions, below are some minor comments:
Abstract: Line 26: Please use “”transverse relaxation (T2)” rather than just T2 since it is the first time the authors mention about this term.
Introduction part:
- A simple scheme of how the SPION works on the MRI application will be helpful for reader to understand, especially about: “longitudinal (T1) or transverse (T2) relaxation”. It should also clearly state: The resolution of MRI is determined by the administration of contrast agents, which enhance the image contrast by shortening the longitudinal (T1) and transverse (T2) relaxation times of water protons.
- Some short sentences to introduce about the gadolinium (Gd) complexes (T1 contrast agents) and its disadvantages due to health risks to patients with kidney and liver diseases will also increase the impact of using T2 contrast agent (e.g., superparamagnetic iron oxide nanoparticles).
- The authors also should discuss about iron oxide nanoparticles with low dimensions, such as ultrathin nanowires (diameter < 3 nm) and ultrasmall nanospheres (<4 nm), which has been increasingly studied as clinically-preferred T1 contrast agents. The nanoparticle size in this work is however >= 5nm, and is used as T2 contrast agent.
Results:
- XRD analysis, please add the 2 standard peak sets for γ-Fe2O3 and Fe3O4 underneath the curve.
- XPS showing the peak at near 735 eV, which normally should just appear in case of Fe3O4. Please compare with the ref: CrystEngComm, 2013, 15, 8166–8172.
- Raman spectra should help on distinguishing between γ -Fe2O3 and Fe3O4.
